# Inter-rating reliability of the Swiss easy-read integrated palliative care outcome scale for people with dementia

Frank Spichiger [1,2]*, Thomas Volken[3], Philip Larkin[1,4], André Anton Meichtry[5], Andrea Koppitz[2]

1 UNIL, Institute of Higher Education and Research in Healthcare, Lausanne, Switzerland, 2 HES-So, School of Health Sciences Fribourg, Switzerland, 3 ZHAW, School of Health Sciences, Winterthur, Switzerland, 4 Palliative and Supportive Care Service, Lausanne University Hospital, Lausanne, Switzerland, 5 School of Health Professionals, Bern University of Applied Sciences, Bern, Switzerland

* frank.spichiger@hefr.ch

**Data Availability Statement:** The full dataset and R-Code to reproduce the data are available at: Frank Spichiger, & Andrea Koppitz. (2023). Inter-rating reliability of the Swiss Easy-Read Integrated

## Abstract

### Background

The Integrated Palliative Care Outcome Scale for People with Dementia is a promising instrument for nursing home quality improvement and research in dementia care. It enables frontline staff in nursing homes to understand and rate the needs and concerns of people with dementia. We recently adapted the measure to include easy language for users from various educational backgrounds.

### Objectives

In this study, we examine the inter-rating reliability of the Integrated Palliative Care Outcome Scale for People with Dementia for frontline staff in nursing homes.

### Methods

In this secondary analysis of an experimental study, 317 frontline staff members in 23 Swiss nursing homes assessed 240 people with dementia from a convenience sample. Reliability for individual items was computed using Fleiss Kappa. Because of the nested nature of the primary data, a generalisability and dependability study was performed for an experimental IPOS-Dem sum score.

### Results

The individual Integrated Palliative Care Outcome Scale for People with Dementia items showed kappa values between .38 (95% CI .3–.48) and .15 (95% CI .08–.22). For the experimental IPOS-Dem sum score, a dependability index of .57 was found. The different ratings and time between ratings explain less than 2% of the variance in the sum score. The different nursing homes make up 12% and the people with dementia make up 43% of the sum score variance. The dependability study indicates that an experimental IPOS-Dem sum score could be acceptable for research by averaging two ratings.

Palliative Care Outcome Scale for People with Dementia [Data set]. Zenodo. https://doi.org/10.5281/zenodo.8036812.

**Funding:** This work was supported by the Swiss Academy of Medical Sciences (https://www.samw.ch/de.html), the Gottlieb and Julia Bangert Foundation (grant number PC 20/17), and the Swiss Academy for Socratic Care (https://www.maeeutik-schweiz.ch/). The funders had no role in study design, data collection and analysis, decision to publish, or preparation of the manuscript.

**Competing interests:** AK, FS, PL and TV are the translators/developers of the Swiss easy-read IPOS-Dem. The Swiss easy-read IPOS- Dem is a secondary outcome measure in a trial where AK is the Principal Investigator. This does not alter our adherence to PLOS ONE policies on sharing data and materials.

## Conclusion

Limited research has been conducted on the measurement error and reliability of patient-centred outcome measures for people with dementia who are living in nursing homes. The Swiss Easy-Read IPOS-Dem is a promising instrument but requires further improvement to be reliable for research or decision making. Future studies may look at its measurement properties for different rater populations or at different stages of dementia. Furthermore, there is a need to establish the construct validity and internal consistency of the easy-read IPOS-Dem.

## Background

*Dementia* is a name given to a group of progressive cognitive diseases [1]. People with dementia may develop impaired functioning, memory, cognition and performance of activities of daily living [1]. According to Sleeman et al. [2], people with moderate to severe dementia face the prospect of health-related suffering. Evidence indicates that people with dementia have inadequate access to the palliative care required for their complex symptoms [2–5]. The complexity of caring for people with dementia arises from their multidimensional symptoms that influence their health; these symptoms also limit accurate prognostic assertions, palliation and treatment [1, 6–8]. In addition, the quality of life and care of people with dementia are also frequently impacted by compromised verbal communication [5, 9–11]. A structured, systematic symptoms assessment process that fosters communication among people with dementia, their family members and frontline staff may help identify symptoms, enable family members to gain insights into caring for people with dementia and improve therapy regimes [12–15]

In Switzerland, people with dementia live in nursing homes for an average of two years and often have multiple comorbidities [16], along with the main diagnosis of moderate to advanced dementia. Swiss nursing homes' usual care follows routinely used assessment instruments [17], namely the Resident Assessment Instrument (RAI-NH), 'Bewohner/-innen-Einstufungs-und-Abrechnungssystem' (BESA). Evaluations using these standardised instruments routinely occur only every six months. Frontline staff in Swiss nursing homes may not have the optimal skills to meet all the care needs of people with Dementia nor are there enough qualified staff [18]. Moreover, limited resources are available for frontline staffs support in Swiss nursing homes, resulting in a lack of systematic use of expertise, assessment instruments and evidence in everyday dementia care [19].

The Integrated Palliative Care Outcome Scale for People with Dementia (IPOS-Dem) is a tool used to inform assessments. The IPOS-Dem is multidimensional; using a person-centred approach, it asks about the most important symptoms and concerns of people with dementia. Using this instrument, frontline staff and family members can identify and address symptoms and concerns [15]. Being attentive to symptoms and concerns is considered a core process in dementia care [20]. The IPOS-Dem may also improve screening, communication, care quality and outcomes in routine care [15]. The IPOS-Dem and its family of tools are informed by empirical qualitative and quantitative work among various populations with palliative care needs [21, 22], and all versions can be downloaded at https://pos-pal.org/.

Thus far, no reliability data have been published for the IPOS-Dem [15, 23, 24]. Ellis-Smith et al. reported on feasibility, mechanisms of action and content validity after analysing focus group and semistructured interview data using directed content analysis [15, 22].

The original IPOS for general palliative care populations, from which the IPOS-Dem is derived, showed inter-rater reliability for 11 of 17 items, ranging from $\kappa_w = .4$ to $\kappa_w = .82$. Several items—including 'Having had enough information', 'Having had practical matters addressed', 'Sharing feelings with family or friends', 'Drowsiness', 'Inner peace' and 'Dry or sore mouth'—repeatedly stood out in analyses, with the $\kappa_w$ ranging between .02 and .29 [22].

The rater population—frontline staff working with people with dementia—is primarily made up of nurses with secondary vocational training degrees or without formal training but with several years of employment and clinical exposure [18, 25]. In Swiss nursing homes, less than one-fifth of the staff working with people with dementia are registered nurses; therefore, we included interns, healthcare assistants and nurses with secondary vocational training.

We developed a Swiss easy-read version of the IPOS-Dem [26] to use in the IPOS-Dem project, which has a stepped-wedge controlled randomised trial (SW-CRT) design [27]. Compared with its predecessor, the easy-read IPOS-Dem is more understandable and adapted to the skill-grade mix and competence of frontline staff in nursing homes [26]. The translation and adaption to IPOS-Dem is described in detail in another study [26]. Here, we present the inter-rating reliability, generalisability and decision study for the easy-read IPOS-Dem, as assessed by frontline staff. Aspects of the validity of the IPOS-Dem will be reported separately to follow Kottner et al.'s [28] Guidelines for Reporting Reliability and Agreement Studies (GRRAS).

## Methods

This is a secondary analysis of a multicentre experimental study with a total of 15 time-shifted assessment periods. For the analysis presented in the present study, data from the baseline measurement period were used. The sample size was determined by power calculations for the overarching SW-CRT, in which the IPOS-Dem was applied. The psychometric analysis of IPOS-Dem was preplanned during the SW-CRT preparation. For this SW-CRT, we aimed to enrol 220 people with dementia living in 22 nursing homes [27] between September 2020 and October 2021. Regarding the raters, we aimed to enrol 20 frontline staff members per nursing home, resulting in a rater population of 440 people. The sample of people with dementia was determined by the nursing homes and based on the agreement of people with dementia to participate (i.e., a convenience sample). The raters were also assigned according to convenience; therefore, no comparison among different levels of training or experience was undertaken. The detailed recruitment process is described in the SW-CRT protocol cited above.

### Ethical approval and consent to participate

The study was approved by the Research Ethics Committee of the canton of Zurich, Switzerland (BASEC-ID: 2019–01847) and was conducted in line with the principles of the Helsinki Declaration [29]. The overarching trial was registered with DRKS00022339. All participants and/or their respective attorneys signed written informed consent for participation and (as outlined in the PLOS consent form) publication. All raters have signed written informed consent for participation and (as outlined in the PLOS consent form) publication.

### Population

**People with dementia.**   People with dementia were included if they (a) were not hospitalised at baseline and, therefore, were physically present in the nursing home at the

commencement of the study, (b1) had a diagnosis of vascular dementia or Alzheimer's disease or (b2) had minimum data sets (MDS) data indicating symptoms of dementia.

**Frontline staff.**   Frontline staff members were invited to participate if they (a) were at least 18 years old, (b) had a tenure of at least 3 months in the nursing home, (c) worked at least 20% of the full-time equivalent, provided continuing care to people with dementia (d) and were able to communicate in German.

## Data collection

Each participating nursing home was assigned a clinical champion, that is, a full-time on-site employee who oversaw recruiting, data collection and the general study coordination with the study team, as outlined in the overarching SW-CRT protocol [27]. At baseline, the clinical champions entered the demographic and clinical details of the people with dementia, as derived from their nursing homes' MDS [30, 31], into our research electronic data capture (REDCap) data management system [32]. A survey developed for the frontline staff was completed by them directly following a training session. The participating staff had 120 minutes of on-site introductory training, and they attempted to complete an assessment for a chosen case using the IPOS-Dem.

Frontline staff were explicitly informed during the training—through an informed consent discussion and written material—that inter-rating agreement was being assessed at baseline. For the reliability study, staff independently assessed people with dementia during the baseline period of 30 days. There were no data captured on which of the staff members submitted the IPOS-Dem to the clinical champion. The clinical champion, however, assured that two independent staff members assessed IPOS-Dem independently during baseline. Staff independently rated and completed the instruments for people with dementia between August 2020 and January 2022. Staff were never blinded to clinical information about the people with dementia and completed the paper version of the IPOS-Dem. The data were subsequently entered into REDCap [32], browser-based software that could give continuous feedback to the clinical champion entering the data (e.g., erroneous or missing data). Automated tests run by REDCap also checked the data for plausibility and completeness.

**Study measures.**   The Swiss easy-read version of the IPOS-Dem consists of 27 items related to physical, psychological, spiritual and practical concerns [26]. While mostly taking a self-proxy perspective [33], it asks three types of questions. After an introduction, there are three open questions about main issues during the last week the person with dementia had from the person with dementia's, the frontline staff's and the family member perspective. Following the textboxes, the user is asked to rate a 19-item list of symptoms and concerns regarding how much the symptoms and concerns impacted the person with dementia during the last week, in their opinion. These items are scored on a 5-point scale ranging from 0 (not at all) to 4 (very severe), with each point having its own descriptor. The symptom list continues with eight more questions, switching to a proxy–proxy perspective by asking how frequently a situation occurred. These items are scored on a 5-point scale ranging from 0 (not at all) to 4 (always), with each point again having its own descriptor. IPOS-Dem closes with three scoreable 'wild card' symptom fields. The IPOS-Dem was completed independently by frontline staff at the baseline of a cluster-randomised trial. The clinical champions oversaw frontline staff members' independent completion of two assessments per person with dementia at baseline. In previous studies [15], it took frontline staff on average between 4 and 12 minutes to complete IPOS-Dem, depending on their experience with the instrument.

People with dementia's sociodemographic information was captured by the clinical champion at baseline, as derived from the nursing home minimum datasets and charts at the time

point. The minimum datasets in Swiss nursing homes we referred to are a translation of the RAI-NH [30] or BESA [31]. The extracted chart and minimum dataset data were gender, marital status, nursing home, dementia type (if diagnosed) and dementia severity (if diagnosed).

## Analysis

For each rating, an experimental IPOS-Dem sum score was calculated by adding the individual item responses of the 27 standard items. The scores are added with list-wise deletions of missing and 'do not know' responses. To inform the analyses of inter-rating reliability, we calculated information on the duration between the two IPOS-Dem assessments at baseline and developed an experimental sum score. The sum score was computed per assessment, with the list-wise exclusion of missing or 'do not know' ratings. The answer option 'do not know' was handled as missing. If not stated otherwise, missing data were excluded pairwise from the item-wise analyses. Sociodemographic and clinical data were analysed for the frontline staff, as well as the people with dementia using frequencies, proportions, ranges and distributions, both per nursing home and in total, with the tidyverse package 1.3.2 for R 4.1.2 [34, 35]. The IPOS-Dem item scores were described in a similar manner.

**Item-wise analysis of inter-rating reliability.** Fleiss' kappa is an extension of Cohen's kappa and can be used for more than two raters [36]; it considers the proportion of agreement beyond chance that would be expected if all ratings had been randomly scored. Fleiss' kappa ranges from 0 to 1, with values closer to 1 indicating higher inter-rater reliability. The coefficient ($\kappa$) is computed by the proportions of expected ($\bar{P}_e$) and observed ($\bar{P}$) agreements between ratings: $\kappa = \frac{\bar{P} - \bar{P}_e}{1 - \bar{P}_e}$. To complement the reporting, the percentage of agreement per item was also calculated and is presented in tables.

**Generalisability study.** Generalisability theory allows for the estimation of reliability for various combinations of raters in complex study designs [37]. Our design was based on 460 observations, with four additional factors: 230 people with dementia; 24 different durations between two assessments; 23 clusters and two ratings. This was a nested design, where some factors were nested within levels from other factors. Therefore, the ratings were nested within durations between the two assessments and nursing homes. Furthermore, people with dementia are nested within ratings and nursing homes. The reliability of the experimental IPOS-Dem sum scores is expressed by generalisability coefficients. Like an intraclass correlation coefficient, the generalisability coefficients indicate the reliability of a scale. By estimating variance components, the generalisability coefficients can be calculated. The variance components are estimated using a restricted maximum likelihood approach.

The variance components were estimated with the experimental IPOS-Dem sum score as the outcome variable and each of the factors (person with dementia, rating, cluster and time between assessments) as a random effect. Reliability was then quantified, with the universe score being the expected IPOS-Dem sum score of a person with dementia over the facets of generalisation for rating but fixed for clusters and time between measurements. The index of dependability ($\Phi$) of a single measurement is the ratio of a person with dementias' score variance to the observed score variance.: $\Phi = \frac{\sigma_P^2 + \sigma_C^2 + \sigma_T^2}{\sigma_P^2 + \sigma_C^2 + \sigma_T^2 + \sigma_\varepsilon^2}$. In this model, the index is computed with a formula for consistency rather than agreement. A consistency model was chosen because IPOS-Dem is considered complex and multidimensional; this was also done to adjust for chance agreement. Model fitting and variance component estimation were performed with the lmer package [38] in R [35] 4.1.2.

**Additional analysis and criteria for interpretation.** The dependability index $\Phi$ represents inter-rating reliability for one assessment sum score for a randomly chosen time and

cluster. To compute the reliability of the mean measure of k measurements, we undertook a decision study. This means that the error variance components are divided by k to quantify the reliability of an average sum score over k repetitions. This decision study can help determine how many repetitions (i.e., ratings) would be required to reach an acceptable dependability Φ. For our analysis, this was performed for k = 1, 2, 3, to six repetitions.

For the interpretation of the results, different interpretation criteria were used. The item floor and ceiling effects were interpreted according to the criteria proposed by McHorn and Tarlov [39]. Their defined threshold for such an effect to occur was 15%, that is, the proportion of the sample rated with the lowest (floor) or highest (ceiling) possible score possible. The κ was interpreted according to Fleiss' [40] classification. Fleiss' classification for the interpretation sets only two cut-off values; kappa values below .40 are deemed 'poor', kappa values between .40 and .75 should be considered 'fair to good', and all kappa values above .75 'are deemed excellent' [40]. The G- and D-Study index values can range from 0 to 1 and are interpreted according to Nunnally's proposed criteria [41]. Nunally [41] described coefficients at .7 as 'modest' and sufficient for early stages of research for instrument development.

## Results

### Observations

We analysed data from 257 people who were recruited from 23 nursing homes. On average, frontline staff completed the two IPOS-Dem measures for the inter-rating reliability analysis at baseline of the SW-CRT within 6.1 days (standard deviation [SD] = 7.4). The majority completed both observations within the first week, while some took up to 30 days to complete the repeated assessments. The heterogeneity in the time between the two assessments per nursing home is illustrated in S1 Table.

### Sample characteristics

Table 1 shows the sociodemographic and clinical details of the people with dementia. Because the data were derived from a multicentre trial, we refer the reader to S1 Table for an illustration of the heterogeneity between the nursing homes.

As expected, 79% of the frontline staff were involved in various nursing roles, as shown in Table 2. Interns, therapists, chaplains and others made up 15% of the raters. The mean tenure was 6.5 years. (Please see S1 Table, which illustrates the heterogeneity between the nursing homes.)

### Item characteristics

The item characteristics for the baseline data are presented in Table 3. At baseline, we were able to match between 139 and 239 ratings per item per person with dementia. The items 'Nausea', 'Shortness of breath' and 'Vomiting' showed substantial floor effects, with more than 80% of the answers concentrating on a rating of 0. For the items 'Family anxious or worried', 'Inner peace' and 'Lost interest', frontline staff chose 'Don't know' in more than 29% of the assessments. Additional item characteristics are provided in S2 Table.

**Inter-rating reliability.** In terms of Fleiss' kappa, the values varied between .39 and .15, as shown in Table 4. The proportions of exact agreement varied between 39% and 89.5%.

### Generalisability and decision study for an experimental sum score

We computed matched IPOS-Dem sum scores for 230 people with dementia; further statistics are shown in Table 5 below. The maximum possible sum score was 108, which was not reached in our sample.

**Table 1. Sociodemographic and clinical details of people with dementia.**

| Variable | N (%) | Mean (SD) | Min–Max (Median) |
|---|---|---|---|
| People with Dementia | 257 (100%) | | |
| Gender | | | |
| *Female* | 180 (70%) | | |
| *Male* | 77 (30%) | | |
| Age | | 86 (7.29) | 56–102 (86) |
| Marital Status | | | |
| *Single* | 21 (8%) | | |
| *Married* | 70 (27%) | | |
| *Divorced* | 30 (12%) | | |
| *Widowed* | 136 (53%) | | |
| Area of Residence | | | |
| *Intermediate* | 176 (68%) | | |
| *Rural* | 43 (17%) | | |
| *Urban* | 38 (15%) | | |
| Dementia | | | |
| *Alzheimer's* | 83 (32%) | | |
| *Vascular* | 22 (9%) | | |
| *Other* | 106 (41%) | | |
| *Not formally diagnosed* | 46 (18%) | | |
| Severity | | | |
| *Mild* | 6 (2%) | | |
| *Moderate* | 81 (32%) | | |
| *Advanced* | 86 (34%) | | |
| *Not applicable* | 84 (32%) | | |

We fitted a linear mixed model to the sum score with person, rating, cluster and occasion as random intercepts.

Based on the variance components shown in Table 6 we computed $\Phi = 0.58$ for a single rating on a random day in a random cluster. In addition, we computed $\Phi$ for a mean of k ratings (k = 1, 2, 3, . . ., 6) to identify an acceptable lower bound of reliability for the sum score as shown in Table 7.

Our dependability study indicates that an acceptable sum score above the .7 could be obtained by averaging the sum scores from two ratings.

## Discussion

The present study aimed to assess the reliability of the newly developed, easy-read IPOS-Dem when used by frontline staff in nursing homes. We computed the generalisability coefficient from two ratings of an experimental sum score and the individual Fleiss' kappa for each item. The $\kappa$ of the items was between .38 (95% CI .3–.48) and .15 (95% CI .08–.22), indicating 'poor' agreement ($\kappa < .4$) when interpreted with Fleiss [40] criteria. An experimental IPOS-Dem sum score was used to enable the computation of a reliability coefficient under the generalisability framework. The findings of these analyses show a G-coefficient of .58. Our decision study shows that, by averaging two ratings, acceptable reliability for research could be obtained. The generalisability study also showed that the differences between participating nursing homes could explain 12% of the variance in the sum IPOS-Dem scores. Only small

**Table 2. Sociodemographic details of frontline staff (i.e., raters).**

| Variable | N (%) | Mean (SD) | Min–Max (Median) |
|---|---|---|---|
| Staff | 311 (100%) | | |
| Age | 304 (98%) | 43 (13.6) | 18–70 (45) |
| Gender | | | |
| *Female* | 277 (89%) | | |
| *Male* | 34 (11%) | | |
| Tenure (years) | | 6.6 (6.6) | 0–32 (5) |
| Occupation | | | |
| *Registered nurse* | 108 (35%) | | |
| *Nursing associate professionals* | 58 (19%) | | |
| *Health care assistants* | 96 (31%) | | |
| *Registered nurse (intern)* | 9 (3%) | | |
| *Nursing associate professionals (intern)* | 19 (6%) | | |
| *Intern* | 1 (< 1%) | | |
| *Other*[a] | 17 (5%) | | |
| *Missing* | 3 (< 1%) | | |
| Education | | | |
| *Tertiary* | 121 (39%) | | |
| *Upper secondary* | 137 (44%) | | |
| *Lower secondary* | 23 (7%) | | |
| *Other* | 28 (9%) | | |
| *Missing* | 2 (< 1%) | | |

[a] 'Other' included: housekeeping staff, chaplains, volunteers and social workers

fractions of the variance were explained by ratings or time between assessments alone. The high proportion of IPOS-Dem sum score variance (41%) explained by residual variance may indicate interactions and measurement errors that must be investigated in future studies. Furthermore, without further investigation into the validity of the IPOS-Dem, the construction of a sum score remains experimental.

## Limitations and strengths

We were able to obtain data from a considerable sample of people with dementia and involve frontline staff with different backgrounds, experiences and education in the primary study. This is the first study to evaluate the psychometric properties of the IPOS-Dem in a larger sample.

The present study has several limitations that we want to highlight. First, we were not able to ensure blinding of the raters regarding prior findings, clinical information and the accepted reference standard measurements like the RAI MDS. Second, there is no consensus in the literature on the stability of the IPOS-Dem ratings, as well as the symptoms and concerns of people with dementia in general. Because routine measurement is undertaken every six months, the relatively research-inexperienced setting and the design of the overarching SW-CRT, we considered one month suitable. We could have determined the sample size based on acceptable CIs (i.e. ± 0.1/ ±0.2) for ICCs reported in previous IPOS studies presented above [42, 43]. With 256 people with dementia, however, we exceeded the typical recommended number of participants in reliability studies often based on rule of thumb (n = 50) [42]; the 95% CIs around the Fleiss kappa are provided in Table 5.

**Table 3. Easy-read IPOS-Dem item characteristics.**

| Item | Mean Score | Score (SD) | None (%) | Some (%) | Moderate (%) | Severe (%) | Very Severe (%) | Don't Know (%) | N Matched Cases |
|---|---|---|---|---|---|---|---|---|---|
| Pain[a] | 1.3 | 1.1 | 26.7 | 35.1 | 25.1 | 10 | 3.1 | 10.6 | 220 |
| Shortness of breath[a] | 0.2 | 0.6 | 83.5 | 10.8 | 4.5 | 0.8 | 0.4 | 5.7 | 232 |
| Weakness[a] | 1.5 | 1.1 | 20.9 | 33.3 | 29.8 | 11.2 | 4.9 | 4.9 | 234 |
| Nausea[a] | 0.2 | 0.6 | 83.9 | 10.7 | 3.8 | 1 | 0.6 | 11.4 | 218 |
| Vomiting[a] | 0.1 | 0.4 | 93.8 | 3.9 | 1.2 | 0.8 | 0.2 | 7.3 | 228 |
| Poor appetite[a] | 0.8 | 1 | 53.7 | 24.7 | 14.7 | 4.5 | 2.4 | 6.5 | 230 |
| Constipation[a] | 0.7 | 0.9 | 56.9 | 25.6 | 13.5 | 3 | 1.1 | 14.6 | 210 |
| Sore or dry mouth[a] | 0.4 | 0.9 | 75.9 | 12.4 | 8.3 | 1.5 | 2 | 18.7 | 200 |
| Drowsiness[a] | 1.5 | 1.1 | 21 | 31.4 | 28.8 | 12 | 6.7 | 6.5 | 230 |
| Poor mobility[a] | 1.2 | 1.4 | 43.6 | 22.2 | 12.5 | 11.3 | 10.3 | 4.5 | 235 |
| Sleeping problems[a] | 0.8 | 1 | 51.9 | 26.2 | 12.8 | 7.5 | 1.7 | 11.4 | 218 |
| Diarrhoea[a] | 0.3 | 0.7 | 78.4 | 14.3 | 5 | 1.5 | 0.8 | 11.8 | 217 |
| Dental problems[a] | 0.6 | 1 | 70.2 | 14.5 | 8.7 | 3.2 | 3.4 | 14.2 | 211 |
| Swallowing problems[a] | 0.5 | 1 | 74.5 | 13.1 | 6.3 | 2.9 | 3.3 | 6.5 | 230 |
| Skin breakdown[a] | 1 | 1.1 | 44.8 | 27.1 | 16.6 | 8.9 | 2.6 | 4.1 | 236 |
| Difficulty communicating[a] | 1.6 | 1.4 | 31.7 | 19 | 22.2 | 14.1 | 13.1 | 4.1 | 236 |
| Hallucinations and/or delusions[a] | 0.8 | 1.1 | 61 | 16.5 | 11.2 | 7.9 | 3.3 | 19.9 | 197 |
| Agitation[a] | 1.7 | 1.3 | 22.6 | 21.8 | 27.7 | 16 | 11.8 | 2.8 | 239 |
| Wandering[a] | 1.3 | 1.4 | 45.3 | 15.1 | 17.1 | 12.2 | 10.2 | 6.5 | 230 |
| Anxious or worried[a] | 1.8 | 1.1 | 15.9 | 20.3 | 37.8 | 21.5 | 4.4 | 3.7 | 237 |
| Family anxious or worried[a] | 1.6 | 1.3 | 28.8 | 22.6 | 24.8 | 11.8 | 12 | 43.5 | 139 |
| Felt depressed[a] | 1.5 | 1 | 18 | 29 | 36.4 | 14.8 | 1.9 | 13 | 214 |
| Lost interest[a] | 1.1 | 1.2 | 42.7 | 22.3 | 19.5 | 12.3 | 3.2 | 31.3 | 169 |
| Inner peace[a] | 1.5 | 0.9 | 9.7 | 48.2 | 25.8 | 13.4 | 3 | 29.3 | 174 |
| Able to interact[a] | 1.5 | 1.3 | 29.5 | 24.9 | 19.9 | 18.9 | 6.8 | 3.3 | 238 |
| Irritable or aggressive[a] | 1.3 | 1 | 27.6 | 26 | 33.2 | 12.1 | 1.2 | 3.7 | 237 |
| Practical matters[a] | 1.4 | 1.1 | 24.5 | 34.3 | 24.9 | 10.6 | 5.7 | 14.6 | 210 |

Item characteristics for baseline data. Items are ordered as they occur in the easy-read IPOS-Dem.

[a]Items with floor effect (more than 15% of answers in lowest category).

The assignment of assessors to people with dementia was delegated to clinical champions, and the assessors' skills and grades were not linked to their respective ratings. The sample of people with dementia was rather heterogeneous, with a fifth lacking a formal diagnosis and different stages of reported severity. The lack of a severity assignment in a third of the sample deterred us from analysing the subsamples of the population and may also have contributed to the observed reliability. To control for the lack of assessment, the use of dementia staging instruments like FAST [44] at the baseline of research projects is highly recommended instead of relying on routine data. These shortcomings of the reported design may contribute to a major part of the unexplained variability in the sum scores.

## Comparison with other instruments for people with dementia

QUALIDEM [45] was developed for observation-based quality of life assessment in people with dementia living in nursing homes. Ettema et al. [45] developed a scale for rating by nursing assistants, placing their scale within a similar scope as the IPOS-Dem. In their study, 68

**Table 4. Item-wise reliability coefficients and proportions of agreement.**

| Item | Kappa | CI Lower Bound | CI Upper Bound | Don't Know | Agreement | Adjacent | Two Scores Apart | Three Scores Apart | Four Scores Apart |
|---|---|---|---|---|---|---|---|---|---|
| Pain[a] | 0.33 | 0.25 | 0.41 | 10.6 | 50.7 | 38.5 | 9.5 | 1.4 | 0 |
| Shortness of breath[a] | 0.35 | 0.25 | 0.45 | 5.7 | 80.6 | 15.1 | 3.4 | 0.4 | 0.4 |
| Weakness[a] | 0.15 | 0.08 | 0.22 | 4.9 | 37.2 | 50.9 | 9.8 | 1.7 | 0.4 |
| Nausea[a] | 0.39 | 0.28 | 0.49 | 11.4 | 80.9 | 14.5 | 3.2 | 0.9 | 0.5 |
| Vomiting[a] | 0.21 | 0.11 | 0.31 | 7.3 | 89.5 | 6.1 | 2.2 | 1.8 | 0.4 |
| Poor appetite[a] | 0.25 | 0.17 | 0.33 | 6.5 | 52.8 | 33.3 | 10.4 | 2.6 | 0.9 |
| Constipation[a] | 0.28 | 0.19 | 0.37 | 14.6 | 56.2 | 29.5 | 10.5 | 3.8 | 0 |
| Sore or dry mouth[a] | 0.3 | 0.21 | 0.4 | 18.7 | 72.8 | 14.9 | 8.9 | 2 | 1.5 |
| Drowsiness[a] | 0.22 | 0.15 | 0.29 | 6.5 | 40.7 | 45.9 | 10.8 | 2.2 | 0.4 |
| Poor mobility[a] | 0.29 | 0.22 | 0.37 | 4.5 | 49.4 | 32.3 | 13.2 | 4.3 | 0.9 |
| Sleeping Problems[a] | 0.28 | 0.2 | 0.37 | 11.4 | 54.3 | 35.6 | 9.1 | 0.5 | 0.5 |
| Diarrhoea[a] | 0.34 | 0.24 | 0.44 | 11.8 | 74.3 | 20.2 | 2.8 | 2.3 | 0.5 |
| Dental Problems[a] | 0.39 | 0.3 | 0.48 | 14.2 | 71.8 | 19.7 | 5.6 | 2.3 | 0.5 |
| Swallowing problems[a] | 0.31 | 0.23 | 0.4 | 6.5 | 71.3 | 18.3 | 7 | 0.9 | 2.6 |
| Skin breakdown[a] | 0.28 | 0.2 | 0.35 | 4.1 | 50 | 32.2 | 12.7 | 4.2 | 0.8 |
| Difficulty communicating[a] | 0.31 | 0.24 | 0.37 | 4.1 | 46.2 | 30.1 | 17.8 | 5.1 | 0.8 |
| Hallucinations and/or delusions[a] | 0.34 | 0.25 | 0.42 | 19.9 | 61.6 | 18.7 | 13.8 | 3.9 | 2 |
| Agitation[a] | 0.26 | 0.19 | 0.32 | 2.8 | 41.8 | 37.2 | 17.6 | 2.9 | 0.4 |
| Wandering[a] | 0.33 | 0.26 | 0.41 | 6.5 | 52.4 | 24.2 | 16.5 | 5.6 | 1.3 |
| Anxious or worried[a] | 0.21 | 0.14 | 0.28 | 3.7 | 40.9 | 46 | 11.4 | 1.3 | 0.4 |
| Family anxious or worried[a] | 0.24 | 0.16 | 0.33 | 43.5 | 41.3 | 29.7 | 20 | 5.8 | 3.2 |
| Felt Depressed[a] | 0.24 | 0.16 | 0.31 | 13 | 44 | 42.1 | 11.1 | 2.3 | 0.5 |
| Lost interest[a] | 0.2 | 0.11 | 0.29 | 31.3 | 44.1 | 30.5 | 16.9 | 7.9 | 0.6 |
| Inner peace[a] | 0.17 | 0.08 | 0.26 | 29.3 | 45.1 | 46.7 | 7.1 | 0.5 | 0.5 |
| Able to interact[a] | 0.27 | 0.2 | 0.34 | 3.3 | 44.1 | 37 | 13.9 | 4.6 | 0.4 |
| Irritable or aggressive[a] | 0.26 | 0.18 | 0.34 | 3.7 | 46 | 42.6 | 10.1 | 1.3 | 0 |
| Practical matters[a] | 0.18 | 0.1 | 0.25 | 14.6 | 39 | 41.3 | 14.1 | 4.7 | 0.9 |

Fleiss' kappa (κ) from two matched independent frontline staff assessments, including 95% confidence intervals (Cis) and proportions of agreement per IPOS-Dem.

Items are ordered as they occur in the easy-read IPOS-Dem.

[a] Items with floor effect (more than 15% of answers in lowest category).

raters assessed 238 people with very severe dementia. Ettema et al. subsequently calculated an overall reliability coefficient between .55 and .79. With later improvements in the German translation of QUALIDEM, reliability coefficients for individual items were improved. This was achieved by increasing the number of response options from four to seven and by the

**Table 5. Characteristics of the IPOS-Dem sum scores for both ratings.**

| Statistics | 1st assessment sum score | 2nd assessment sum score |
|---|---|---|
| Number of matched cases | 230 | 230 |
| Mean (SD) | 25.3 (13.0) | 28 (13.7) |
| Median (IQR) | 25 (17.75) | 27 (17.75) |
| Range (min–max) | 0–73 | 0–78 |

**Table 6. Variance components with respective proportions.**

| Component | Absolute variance component | % variance component |
|---|---|---|
| Person $\sigma_P^2$ | 79.28 | 42.71 |
| Time between ratings $\sigma_T^2$ | 3.25 | 1.75 |
| Rating $\sigma_R^2$ | 3.36 | 1.81 |
| Cluster $\sigma_C^2$ | 22.99 | 12.39 |
| Residuals $\sigma^2$ | 76.73 | 41.34 |

**Table 7. Dependability coefficients for multiple ratings.**

| Number of ratings (k) | Coefficients |
|---|---|
| 1 | 0.579 |
| 2 | 0.733 |
| 3 | 0.805 |
| 4 | 0.846 |
| 5 | 0.873 |
| 6 | 0.892 |

development of a detailed guide booklet [46, 47]. Dichter et al.'s German QUALIDEM study involved 36 people with advanced dementia who were rated by four caregivers with the revised QUALIDEM. In Dichter's QUALIDEM paper, only 6 out of 18 items showed floor or ceiling effects, although the authors opted to define floor effects by mean scores, with kappa values between .31 and .62. The items with the lowest reliability coefficients in the study were from the affect and social subscales. Similarly, some of the items that had low reliability in our study (i.e., 'Felt depressed' or 'Anxious or Worried').

Dichter et al. concluded that the QUALIDEM subscales generally showed sufficient reliability (between .64 and .91). However, in their related work, Dichter et al. [48] highlighted the lack of reliability investigations for instrument translations specific to the dementia population. The current Swiss guideline for dementia care in nursing homes [49] does not include any recommendations for instruments that can be used with all frontline staff members (e.g., Health care assistants, nursing associate professionals and interns).

Other popular instruments used for research on people with dementia are the Quality of Dying Instruments End-of-Life in Dementia Comfort Assessment in Dying (EOLD-CAD) and the Quality of Dying in Long-Term Care (QOD-LTC) [50–52]. However, the EOLD-CAD's reliability coefficient was moderate (0.59) and fair for the QOD-LTC (0.28) [50].

A review of instruments tested in long-term care settings by Ellis-Smith et al. [14] showed that different symptom-specific measures had reliability coefficients ranging between .76 and .73 for pain, .47 and .66 for measures of oral health and .20 for the single identified depression scale. In accordance with Dichter et al. and Kupeli et al. [48, 53], Ellis-Smith et al. highlighted that the evaluation of psychometric properties for many instruments is lacking. The findings regarding measurement properties identified above is in line with Soest-Poortvliet et al. [54], who looked at instruments evaluating end-of-life care and dying in long-term care residents. Their review of different instruments showed reliability coefficients between .25 and .59. These and our findings imply the difficulty [55] and complexity [48, 56] of evaluating patient outcomes in people with dementia.

## Implications

**Clinical practice.**  With the evidence reported in the present, the Swiss Easy-Read IPOS-Dem cannot be recommended for routine use in clinical practice or decision making. Further research into its psychometric properties needs to be conducted. To improve the reliability of the IPOS-Dem, additional actions targeting rating and observation procedures could be proposed. For example, a handbook could complement raters' training; this has already proven to be successful in developing other measures for this population [57, 58]. However, the underlying philosophy of user-friendly symptoms and concerns assessment permeates the IPOS family of measures [22]. An advantage of using the easy-language IPOS-Dem is its accessibility to frontline staff and family members in clinical practice without extensive training or a reading exercise in a handbook. This strength of the IPOS-Dem was theorised as mitigating setting-specific barriers to the effective implementation of palliative and person-centred care, such as high staff turnover, low incentives for professional staff development and the supersaturation of methods and instruments for geriatric care.

**Research.**  With the evidence reported here, the Swiss Easy-Read IPOS-Dem experimental sum score might be used in research when averaged over two ratings. Because of these limitations, we caution against generalising our findings to other populations, settings and configurations of rater populations. Furthermore, the structural validity and validity of the sum score must be investigated first. Future studies investigating the reliability of the easy-read IPOS-Dem may avoid specific sources of variation in the ratings. There are a few options by means of restrictions in the design of such psychometric studies. A classical fully crossed design to determine test–retest and interrater reliability could be realised. First, researchers could restrict the rater population regarding qualifications and clinical exposure in a future study. Second, rigid assessment scheduling could be imposed on the day, the time between assessments and other factors. To date, there has been no guidance on the frequency at which routine assessments of symptoms and concerns in people with dementia should be conducted; therefore, we had no guiding frequency for imposing limitations on the scheduling of assessments or rater–subject assignments. Further improvements and changes regarding implementation and development will be derived from the experience of our colleagues at the United Kingdom Outcomes Assessment and Complexity Collaborative [59] and findings from the Australian Palliative Aged Care Outcomes Collaborative [12].

## Conclusion

Comprehensive studies on the reliability of multidimensional instruments for people with dementia living in nursing homes have been infrequent. Especially in translated measures, reviews have not reported many publications on this measurement property. Generally, the reliability coefficients of most instruments to rate individual symptoms, quality of care or health-related quality of life in people with dementia hover below acceptable thresholds for clinical decision making and research. Some of the easy-read IPOS-Dem items have shown comparably poor coefficients. The experimental IPOS-Dem sum score may be reliable if averaged over two ratings. However, its validity needs to be investigated first. The present study has provided comprehensive information on the statistical parameters of measurement properties in the Swiss easy-read IPOS-Dem for its intended rater population. Our research shows that further development is needed to improve the easy-read IPOS-Dem to the point that the results can be considered reliable for research on caring quality and clinical decision making.

## Supporting information

**S1 Table. Cluster-wise sociodemographic statistics.** This file contains tabular data for each cluster in a long format.
(HTML)

**S2 Table. Item characteristics.** This file shows additional item characteristics for the easy-read IPOS-Dem and complements Table 3.
(HTML)

## Acknowledgments

We would like to thank the frontline staff who were involved in this study. Furthermore, we wish to thank the clinical champions who participated: A. Beqiri, R. Benz, M. Bonaconsa, A. Brunner, A. Conti, M. Deflorin, D. Deubelbeiss, L. Ebener, S. Egger, E. Eichinger, D. Elmer, A. Ermler, M. Fuhrer, C. Grichting, M. Havarneanu, H. Hettich, E. Hoffmann, E. Imgrueth, R. Juchli, I. Juric, K. Knöpfli, S. Kuonen, F. Laich, H. Meiser, N. Mergime, B. Michel, C. Ming, F. Müller, C. Niederer, G. Parkes, P. Piguet, A. Repesa, C. Ritz, B. Santer, A. Schallenberg, C. Schweiger, M. Spitz, and R. Strunck. Also, many thanks to F. Murtagh for discussing the results and IPOS-Dem with us.

## Author Contributions

**Conceptualization:** Frank Spichiger, Thomas Volken, Andrea Koppitz.

**Data curation:** Frank Spichiger.

**Formal analysis:** Frank Spichiger, André Anton Meichtry.

**Funding acquisition:** Andrea Koppitz.

**Investigation:** Frank Spichiger.

**Methodology:** Frank Spichiger, Thomas Volken, Philip Larkin, Andrea Koppitz.

**Project administration:** Andrea Koppitz.

**Resources:** Frank Spichiger, Andrea Koppitz.

**Supervision:** Thomas Volken, Philip Larkin, Andrea Koppitz.

**Validation:** Thomas Volken, André Anton Meichtry, Andrea Koppitz.

**Visualization:** Frank Spichiger.

**Writing – original draft:** Frank Spichiger.

**Writing – review & editing:** Frank Spichiger, Thomas Volken, Philip Larkin, André Anton Meichtry, Andrea Koppitz.

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
