## [Decision Letter · Decision Letter 0]

3 Nov 2022

PONE-D-22-22847Inter-rater and test-retest reliability of the Swiss easy-read Integrated Palliative Care Outcome Scale for People with DementiaPLOS ONE

Dear Dr. Spichiger,

Thank you for submitting your manuscript to PLOS ONE. After careful consideration, we feel that it has merit but does not fully meet PLOS ONE’s publication criteria as it currently stands. Therefore, we invite you to submit a revised version of the manuscript that addresses the points raised during the review process.

We look forward to receiving your revised manuscript.

Kind regards,

Jamie Males

Editorial Office

PLOS ONE

Journal Requirements:

Reviewers' comments:

Reviewer's Responses to Questions

**Comments to the Author**

1. Is the manuscript technically sound, and do the data support the conclusions?

Reviewer #1: Partly

Reviewer #2: Yes

Reviewer #3: No

Reviewer #4: Yes

Reviewer #5: Yes

2. Has the statistical analysis been performed appropriately and rigorously? 

Reviewer #1: Yes

Reviewer #2: I Don't Know

Reviewer #3: Yes

Reviewer #4: I Don't Know

Reviewer #5: Yes

3. Have the authors made all data underlying the findings in their manuscript fully available?

Reviewer #1: Yes

Reviewer #2: Yes

Reviewer #3: No

Reviewer #4: Yes

Reviewer #5: No

4. Is the manuscript presented in an intelligible fashion and written in standard English?

Reviewer #1: Yes

Reviewer #2: Yes

Reviewer #3: No

Reviewer #4: Yes

Reviewer #5: Yes

5. Review Comments to the Author

Reviewer #1: Dear authors,

Thank you very much for the opportunity to review your very precise and well prepared article. It is a very important contribution to this field of assessing the quality of life of people with dementia.

I have minor comments:

Have you tried to assess whether the evaluation of IPOS-dem somehow differ according to profession of rater?

How was the situation when the raters were the same profession?

Could you add the information about how long does it take to complete IPOS-dem? This might be important for clinical practice.

Your sample was heterogeneous in the severity of dementia, have you thought of the possibility that this also could cause the difference? From my perspective it might be easier to evaluate the quality of life of patients with mild dementia who could be able to communicate, rather then in situation of patients with severe dementia. Try to discuss that in Discussion.

Could you elaborate more in Discussion about the items that are not very reliable, such as Loss of interest? Add your interpretation of that.

In your article you are bringing important information, in Conclusion I miss the information about future direction in this field explicitly related to IPOS-dem, what is your opinion about this tool.

Reviewer #2: The authors examined the "Integrated Palliative Care Outcome Scale for People with Dementia" (IPOS Dem) in Swiss nursing homes for inter-rater reliability and additionally examined test-retest reliability after 1, 2 and 3 months. In 23 nursing homes in Switzerland, 317 staff members applied the IPOS Dem over 4 months to a total of 240 persons with dementia. Neither inter-rater nor test-retest reliability could be proven, adjustments to the IPOS Dem are necessary.

The manuscript is well written, follows a concise structure and the tables are easy to understand. As far as I can evaluate this, the English language is sufficient. Nevertheless I would recommend to invite a statistician to review the methods in detail.

Please let me note some questions and concerns:

1) Background, The rater population: I think this should be mentioned in the Methods section.

2) Analysis: What program was used for statistical analysis? And which coefficients were defined as moderate, high and so on? Which coefficients were expected to be satisfactory?

3) For the reader, it is not clear, why higher correlations in test-retest analysis were expected. Isn`t it possible, that things have changed in patient care, which are meant to change, so that the test-retest even should be lower to indicate for these changes?

4) The manuscript is very technical, but the reader misses arguments on the clinical meaning of the results. Also little literature is embedded to discuss the results.

Reviewer #3: Dear authors,

Please find my comments attached as a pdf.

Reviewer #4: In general, medical professions have different educational background and medical knowledge, and it is difficult to share the same language. This is especially true in the field of dementia care. The IPOS-Dem used in this study is expected to be used as a standard scale internationally in the future. This study is significant because the IPOS-Dem, which is usually used by physicians and nurses, was modified so that it can be used by nursing staff as well.

　The results were not sufficiently reliable. This is a well-designed study with a large number of subjects so that this study suggest strong negative evidence. Then, it is expected to be taken to the next step according to these results. However, the following points need to be confirmed

I think the authors should describe the process to modify the scale for easy-to-read, what the authors paid attention to, and how the authors did it, using examples. This would be helpful for future researchers.

What frontline staff is should be clarified; there is a lot of N/A and “Other” in Occupation, What is this?

The biggest concern is that in the statistical analysis, I don't think the Kappa coefficient can be calculated in the standard way since the evaluators are not fixed. It may be difficult to calculate the ICC as well. We need a more detailed presentation of how it was calculated.

How was “do not know” handled in the analysis?

The abstract only states that there was no ICC>0.7 item and based on that the conclusion is that the reliability was not sufficient. While this is honest, ICC >0.5 or higher seems like a good figure for a single item alone in such an assessment of dementia symptoms. Compared to studies of other rating scales, it could be written a little more positively.

What about the reliability of the IPOS-Dem total score? If the reliability of the total score is high, it would be useful as an endpoint as a global measure.

The authors stated that "The ICCs for 2 assessments made a month apart varied between .59 and .18 and increased 243 at both 3 assessments (ICC(2,1) = .72-.37) and 4 assessments (ICC(2,1) = .72-.37)." It would be clearer to compare averages of timepoint. One item has a greater impact on the maximum and minimum. In this analysis, it is vomiting, which is infrequent and marginally biased, so the ICC will also be small.

In the future, it may be possible to narrow down the number of evaluators. Couldn't the analysis be limited to evaluations by registered nurses? The authors would think it is not in the real situation, but, I think registered nurse evaluate and share it with the staffs if I use it in my country.

Why is there no age in Tab 2?

Reviewer #5: This study investigated the reliability (inter-rater and test-retest) of an easy-read version the IPOS-Dem tool for staff-assessment of palliative care outcome in people with dementia. The authors conclude that the reliability of the IPOS-Dem is below acceptable levels.

The report is clearly presented; the study seems to be methodologically sound and the conclusions plausible. The choice of intraclass correlation coefficient (ICC(2,1)) appears to be appropriate.

I am sceptical about the use of a binary ‘nominal score’ to identify patients with ‘no change’ in status between successive assessments. Such cases are then used to calculate test-retest reliability, i.e. the assumption is that these cases should show identical values at each assessment. But since the same rater has to rate the case as ‘change’ or ‘no change’ and simultaneously to rate the current outcome status, the amount of test-retest agreement or disagreement seems to merely reflect the consistency between the change/no-change rating and the current-status rating. A better test-retest measure might be obtained by using and comparing the two baseline values.

Additionally, it would be interesting to see whether inter-rater differences could be partly attributed to consistent differences between types of assessors, e.g. between registered nurses, assistants and interns, and to present the reliability measures (inter-rater and test-retest) within these subgroups. Might it be true that higher qualified staff give more reliable assessments?

Minor points

Line 182: please explain the subscripts r, c and rc in the formula. Presumably these refer to rows and columns, but what do rows and columns represent in this case?

Table 3: what is meant by occupation ‘not applicable’ (22%)?

Line 242-3: it is unclear what is meant by 'at 3 assessments': means looking at discrepancies between assessment 2 months apart? If so, the tendency for reliability to increase with time apart seems counterintuitive. In Table 6 the ICC values in the 2-month and 3-month columns are identical, with the sole exception of the first row (item ‘Anxious or worried’).

Line 261: What is meant by ‘restriction’ in ‘specific sources of variation in the measurements caused by restriction’?

Concerning data availability: the authors state precisely when and where the complete data will be made availiable.

6. PLOS authors have the option to publish the peer review history of their article (what does this mean?). If published, this will include your full peer review and any attached files.

Reviewer #1: No

Reviewer #2: No

Reviewer #3: **Yes: **Christina Ramsenthaler

Reviewer #4: No

Reviewer #5: **Yes: **Jeremy Franklin

---

## [Author Response · Author response to Decision Letter 0]

15 Apr 2023

Reviewer #1: Dear authors,

Thank you very much for the opportunity to review your very precise and well prepared article. It is a very important contribution to this field of assessing the quality of life of people with dementia.

I have minor comments:

> Thank you for your review. It is appreciated and we have taken your feedback into account in revising our manuscript.

Have you tried to assess whether the evaluation of IPOS-dem somehow differ according to profession of rater?

How was the situation when the raters were the same profession?

> We agree that the qualification of the assessors would be an important component within the study. We have acknowledged this and suggested it as a topic for further research in the Limitations section. Further, the revised manuscript now explicitly states the consequence of convenience sampling. We further acknowledge the lack of data regarding which staff member assessed which person with dementia as a major limitation of the study.

Could you add the information about how long does it take to complete IPOS-dem? This might be important for clinical practice.

> Thank you for this important comment. We have added this to the technical instrument description in the methods section. According to Ellis-Smith the mean time it took to complete IPOS-Dem at baseline was 8.48 minutes (SD 4.98) and at their final time point, 5.60 minutes (SD 1.45). 

Your sample was heterogeneous in the severity of dementia, have you thought of the possibility that this also could cause the difference? From my perspective it might be easier to evaluate the quality of life of patients with mild dementia who could be able to communicate, rather then in situation of patients with severe dementia. Try to discuss that in Discussion.

> The point you raise is now discussed in the last paragraph of the strengths and limitations section of the revised manuscript. We refrained from relying on the staging documented in the health records. Future psychometric studies should assess dementia stage as part of their baseline assessment. 

Could you elaborate more in Discussion about the items that are not very reliable, such as Loss of interest? Add your interpretation of that.

> We have completely reworked the discussion in accordance with the reviewer suggestions. We think this "per-item" discussion would only be viable in a study with a fully crossed design where more factors have been controlled for.

In your article you are bringing important information, in Conclusion I miss the information about future direction in this field explicitly related to IPOS-dem, what is your opinion about this tool.

> The conclusion was reformulated to clarify the need for further development and psychometric evaluation. 

---

Reviewer #2: The authors examined the "Integrated Palliative Care Outcome Scale for People with Dementia" (IPOS Dem) in Swiss nursing homes for inter-rater reliability and additionally examined test-retest reliability after 1, 2 and 3 months. In 23 nursing homes in Switzerland, 317 staff members applied the IPOS Dem over 4 months to a total of 240 persons with dementia. Neither inter-rater nor test-retest reliability could be proven, adjustments to the IPOS Dem are necessary.

The manuscript is well written, follows a concise structure and the tables are easy to understand. As far as I can evaluate this, the English language is sufficient. Nevertheless I would recommend to invite a statistician to review the methods in detail.

Please let me note some questions and concerns:

> Thank you for your review and summary. The overall structure of the manuscript was changed to only report on the baseline data due to the complexity of the design. Two statisticians, one for overall supervision and in co-analysis with the first author were involved in this revised manuscript. We improved the quality and the veracity of the paper.

1) Background, The rater population: I think this should be mentioned in the Methods section.

> We agree and also acknowledge from the other reviewers comments that a more in-depth description for the raters/frontline staff is needed in the methods and results sections. A brief description in the introduction section of the „target population“ for this „PROM“ is however indicated both - by the GRASS guidance and the COSMIN reporting recommendations (I2). Therefore, this brief description providing additional context for the setting and so has retained in the revision.

2) Analysis: What program was used for statistical analysis? And which coefficients were defined as moderate, high and so on? Which coefficients were expected to be satisfactory?

> We agree and also acknowledge from the other reviewers comments that a more in-depth description for the raters/frontline staff is needed in the methods and results sections. A brief description in the introduction section of the „target population“ for this „PROM“ is however indicated both - by the GRASS guidance and the COSMIN reporting recommendations (I2). Therefore, this brief description providing additional context for the setting and so has retained in the revision.

3) For the reader, it is not clear, why higher correlations in test-retest analysis were expected. Isn`t it possible, that things have changed in patient care, which are meant to change, so that the test-retest even should be lower to indicate for these changes?

> The test-retest section of the paper was removed.

4) The manuscript is very technical, but the reader misses arguments on the clinical meaning of the results. Also little literature is embedded to discuss the results.

> We have completely reworked the discussion in accordance with the reviewer suggestions

---

Reviewer #3: Dear authors,

Please find my comments attached as a pdf.

> Please find our responses in the attached html point-by point response.

---

Reviewer #4: In general, medical professions have different educational background and medical knowledge, and it is difficult to share the same language. This is especially true in the field of dementia care. The IPOS-Dem used in this study is expected to be used as a standard scale internationally in the future. This study is significant because the IPOS-Dem, which is usually used by physicians and nurses, was modified so that it can be used by nursing staff as well.

　The results were not sufficiently reliable. This is a well-designed study with a large number of subjects so that this study suggest strong negative evidence. Then, it is expected to be taken to the next step according to these results. However, the following points need to be confirmed

> Thank you for your review and summary. You will find our point-by point response below. 

I think the authors should describe the process to modify the scale for easy-to-read, what the authors paid attention to, and how the authors did it, using examples. This would be helpful for future researchers.

> Thank you and we agree. The translation and adaption is an extensive and laborious process. In this paper we focus on other psychometric properties of the instrument. 

A very detailed explication of our changes to the instrument is published in https://doi.org/10.1186/s41687-022-00420-7 [26]. This is now stated explicitly in the revised manuscript.

What frontline staff is should be clarified; there is a lot of N/A and “Other” in Occupation, What is this?

> We re-analyzed the sociodemographic dataset for the staff; There was an issue with categorising/naming. Care role names are not protected by law standardised in Switzerland and so for vocational and staff members in nursing homes without professional qualifications, there is great heterogeneity. Also the additional vocationally trained tiers of care staff has introduced further confusion. NA was shorthand for these, nursing associate professionals. Truly left blank were only three of these surveys occupation item. In the revised manuscript the role designations now adhere to OECD standards. For the option „other“ mostly Health care assistants chose to fill in a free text field which we were now able to recategorise appropriately. Furthermore some RN chose to provide leadership or clinical trainer roles. To better illustrate other a footnote has been added to the table.

The biggest concern is that in the statistical analysis, I don't think the Kappa coefficient can be calculated in the standard way since the evaluators are not fixed. It may be difficult to calculate the ICC as well. We need a more detailed presentation of how it was calculated.

> This is a valid propostion. ICC (2,1) might be able to capture the complexity of the situation to some degree, but since the assumption of interval-scaled scores is not fullfilled, Kappa only is featured in the revised manuscript. After discussion and further review of the literature, we conclude that Fleiss Kappa fits the design and can be used for item wise analysis of our scores.

How was “do not know” handled in the analysis?

> Thank you for highlighting this, we have added a sentence to clarify the issue.

The abstract only states that there was no ICC>0.7 item and based on that the conclusion is that the reliability was not sufficient. While this is honest, ICC >0.5 or higher seems like a good figure for a single item alone in such an assessment of dementia symptoms. Compared to studies of other rating scales, it could be written a little more positively.

> Whith a nuanced reporting (range of achieved coefficients) the reporting of the reliability coefficients in the abstract is more precise.

What about the reliability of the IPOS-Dem total score? If the reliability of the total score is high, it would be useful as an endpoint as a global measure.

> Thank you. A G-Study and D-Study for an experimental total score have been undertaken and added to the manuscript. 

The authors stated that "The ICCs for 2 assessments made a month apart varied between .59 and .18 and increased 243 at both 3 assessments (ICC(2,1) = .72-.37) and 4 assessments (ICC(2,1) = .72-.37)." It would be clearer to compare averages of timepoint. One item has a greater impact on the maximum and minimum. In this analysis, it is vomiting, which is infrequent and marginally biased, so the ICC will also be small.

In the future, it may be possible to narrow down the number of evaluators. Couldn't the analysis be limited to evaluations by registered nurses? The authors would think it is not in the real situation, but, I think registered nurse evaluate and share it with the staffs if I use it in my country.

> This is correct. This is discussed in the limitations and recommendations for further research.

Why is there no age in Tab 2?

> Our thanks to reviewer #4 for noting this. We initially planned to report this only in text. However this lost in editing and due to an oversight not added back to the table. The revised manuscript now has the statistics for the age of the people with dementianincluded.

---

Reviewer #5: This study investigated the reliability (inter-rater and test-retest) of an easy-read version the IPOS-Dem tool for staff-assessment of palliative care outcome in people with dementia. The authors conclude that the reliability of the IPOS-Dem is below acceptable levels.

> Thank you for this summary and the additional points you raised in the review. You will find major changes to the manuscript and point-by point responses to your suggestions and questions below. 

The report is clearly presented; the study seems to be methodologically sound and the conclusions plausible. The choice of intraclass correlation coefficient (ICC(2,1)) appears to be appropriate.

I am sceptical about the use of a binary ‘nominal score’ to identify patients with ‘no change’ in status between successive assessments. Such cases are then used to calculate test-retest reliability, i.e. the assumption is that these cases should show identical values at each assessment. But since the same rater has to rate the case as ‘change’ or ‘no change’ and simultaneously to rate the current outcome status, the amount of test-retest agreement or disagreement seems to merely reflect the consistency between the change/no-change rating and the current-status rating. A better test-retest measure might be obtained by using and comparing the two baseline values.

> As previously noted, the test-retest section of the paper has been removed. Therefore we removed the description of global rating of change and time periods. Your suggestion is somewhat reflected in the G-Study. Thank you.

Additionally, it would be interesting to see whether inter-rater differences could be partly attributed to consistent differences between types of assessors, e.g. between registered nurses, assistants and interns, and to present the reliability measures (inter-rater and test-retest) within these subgroups. Might it be true that higher qualified staff give more reliable assessments?

> The assessments are not linked with individual raters, therefore no subgroup analysis of this kind is possible with our data. We tried to reflect your suggestions in the G-Study section of the revised manuscript.

Minor points

Line 182: please explain the subscripts r, c and rc in the formula. Presumably these refer to rows and columns, but what do rows and columns represent in this case?

> The relevant ICC Formula has been removed from the revised paper, since the assumption of interval-scaled scores is not fullfilled.

Table 3: what is meant by occupation ‘not applicable’ (22%)?

> We re-analyzed the sociodemographic dataset for the staff; There was an issue with categorising/naming. Care role names are not protected by law or standardised in Switzerland, for vocational and non-training staff members in nursing homes there is great heterogeneity. Also the additional vocationally trained tiers of care staff has introduced further confusion. NA was shorthand for these, nursing associate professionals. Only three surveys left the occupation item blank. In the revised manuscript the role designations now adhere to OECD standards. For the option „other“ mostly Health care assistants chose to fill in a free text field which we were now able to recategorise appropriately. Furthermore some RN chose to provide leadership or clinical trainer roles. To better illustrate other a footnote has been added to the table.

Line 242-3: it is unclear what is meant by 'at 3 assessments': means looking at discrepancies between assessment 2 months apart? If so, the tendency for reliability to increase with time apart seems counterintuitive. In Table 6 the ICC values in the 2-month and 3-month columns are identical, with the sole exception of the first row (item ‘Anxious or worried’).

> Yes, this was a counterintuitive finding. It is rooted in the way ICC(1,2) was calculated to reflect agreement rather than consistency. The test-retest section of the paper has been removed.

Line 261: What is meant by ‘restriction’ in ‘specific sources of variation in the measurements caused by restriction’?

> Thank you, the sentence you cite was incorrect and has been removed. 

Concerning data availability: the authors state precisely when and where the complete data will be made availiable.

> Thank you, yes the declarations section(s) is/are submitted separately from the manuscript file. 

We do not know how this is presented to the reviewers, since there seems to be some confusion so we hope this is now clarified. 

We note the data availability section here: "After completion of the overarching trial and embargo for publications, the anonymized dataset will be made available at https://doi.org/10.5281/zenodo.4008427 please contact the first author to access the data before the embargo ends. We provide the code for the analysis at https://doi.org/10.5281/zenodo.4008429.". We hope this provides the necessary clarification.

---

## [Editor Report · Decision Letter 1]

19 May 2023

Inter-rating reliability of the Swiss Easy-Read Integrated Palliative Care Outcome Scale for People with Dementia

PONE-D-22-22847R1

Dear Dr. Spichiger,

We’re pleased to inform you that your manuscript has been judged scientifically suitable for publication and will be formally accepted for publication once it meets all outstanding technical requirements.

Kind regards,

Mitsunori Miyashita, R.N. Ph.D.

Guest Editor

PLOS ONE
---

## [Editor Report · Acceptance letter]

24 Jul 2023

PONE-D-22-22847R1 

Inter-rating reliability of the Swiss Easy-Read Integrated Palliative Care Outcome Scale for People with Dementia 

Dear Dr. Spichiger:

I'm pleased to inform you that your manuscript has been deemed suitable for publication in PLOS ONE. Congratulations! Your manuscript is now with our production department. 

Kind regards, 

on behalf of

Dr. Mitsunori Miyashita 

Guest Editor

PLOS ONE